# Adaptive Multi-Prototype Grouping Alignment for Domain Adaptive 3D Detection

## Abstract

3D object detection is crucial for autonomous driving and robotics, but models often perform poorly when deployed in new environments due to domain shifts. While 3D unsupervised domain adaptation methods aim to address this issue, they still struggle with two key challenges: insufficient cross-domain feature alignment and ambiguous foreground-background decision boundaries. In this paper, we propose an Adaptive Multi-Prototype Grouping Alignment framework that addresses these challenges. Our method automatically discovers and dynamically updates feature groups, enabling adaptively cross-domain feature alignment in a fine grained manner. Additionally, we develop two techniques to address the issue of ambiguous foreground-background decision boundaries: Noise Background Hybrid Augmentation that leverages labeled source instances to enhance adaptation in uncertain regions, and Noise Foreground Contrastive Learning that improves foreground-background discrimination by pushing low-confidence features away from prototypes. We conduct extensive experiments on multiple cross-domain benchmarks and the results demonstrate the superiority of our method over the state-of-the-art methods.

## 1 Introduction

As a fundamental task in computer vision, 3D object detection plays a crucial role in robotic navigation and autonomous driving Arnold et al. (2019); Guo et al. (2020); Li et al. (2020); Malavazi et al. (2018); Ahmed et al. (2018). With the development of large-scale annotated datasets and deep learning techniques, this task has achieved remarkable success Lang et al. (2019); Shi et al. (2019; 2023). However, in real-world scenarios, models often suffer from severe performance degradation due to domain shift between the training data and deployment environments Wang et al. (2020). These shifts typically stem from sensor discrepancies, scene diversity, geographical variations, and weather conditions. A straightforward approach would be to annotate large-scale datasets for each target domain, but the high annotation cost makes it impractical for large-scale deployment. Consequently, Unsupervised Domain Adaptation (UDA) on 3D detection Yang et al. (2021; 2022); Hu et al. (2023); Chen et al. (2024a) has become a critical research topic, aiming to align cross-domain feature spaces and transfer knowledge from the source domain to the target domain without requiring target domain labels.

Recently, several self-training 3D UDA Yang et al. (2021); Hu et al. (2023); Chen et al. (2024b) methods have gained attention for their excellent performance. These methods generate pseudo labels for the target domain using pretrained models and perform iterative training to reduce domain shift through implicit feature alignment. Existing research mainly adopts single-factor alignment strategies that focus on specific domain factors, such as object size Wang et al. (2020); Yang et al. (2021) or point cloud density Hu et al. (2023); Wei et al. (2022). However, despite progress in bridging the domain gap, two challenges remain unsolved: (1) **Insufficient cross-domain feature alignment**: Cross-domain feature differences result from multiple factors. Existing single-factor alignment strategies are insufficient to address the distribution differences in the feature space. As shown in Fig. 1a, existing methods often confuse features from different categories in the target domain, leading to performance degradation. Moreover, these methods mainly focus on **single-category 3D UDA** Yang et al. (2019); Shin et al. (2024); Chen et al. (2024b). In more complex and realistic multi-category scenarios Chen et al. (2023), they frequently struggle to maintain good performance due to inter-category feature interference. (2) **Ambiguous foreground-background**

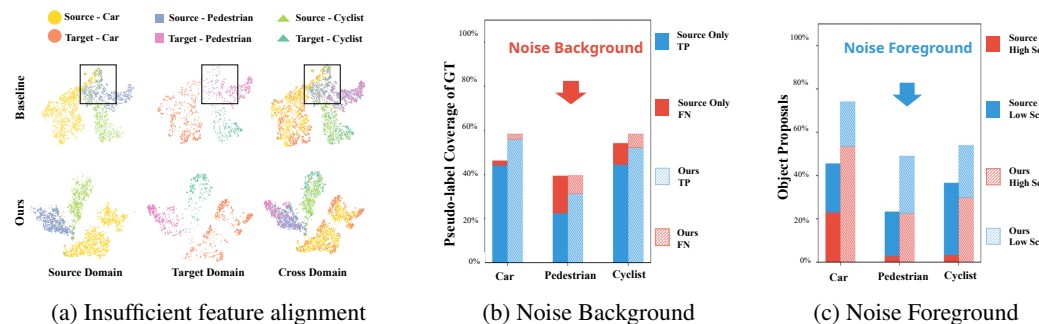

(a) Insufficient feature alignment    (b) Noise Background    (c) Noise Foreground

Figure 1: (a) Visualization of BEV features showing confusion across different categories and scattered distribution patterns in the baseline method Yang et al. (2022); (b) Noise Background: Lots of positive samples are incorrectly labeled as negative in the pseudo-labeling process. (c) Proposals contain substantial noise with low-confidence foreground predictions that are easily confused with background.

**decision boundaries**: Due to the physical limitations of LiDAR, point clouds from distant objects and occluded areas are often sparse, making the decision boundary between foreground and background ambiguous. As shown in Fig. 1b, the high-confidence filtering mechanism of pseudo labels misclassifies many positive samples as negative, leading to the **noisy background** issue. Moreover, as shown in Fig. 1c, many easily confused background proposals are predicted as low-confidence foreground during training, further blurring the decision boundary and leading to the **noisy foreground** issue.

An intuitive solution to the issue of insufficient cross-domain feature alignment is to use prototypes to align cross-domain features, as explored in 2D UDA Liu et al. (2025); Lin et al. (2022). However, we argue that two points need to be considered: (1) Point cloud features are influenced by multiple factors, such as observation angles, distances and etc. These factors cause features to exhibit multi-center distribution characteristics. As shown in Fig.1a, samples of the category appear scattered in the feature space. It is challenging to align features using a universal prototype. (2) In a multi-prototype setting, it is necessary to consider how to assign features to the corresponding prototypes. Although Li et al. (2023) proposes a heuristic grouping strategy based on observation angles, this fixed grouping strategy is suboptimal when facing complex domain differences and struggles to adaptively align transferable features. In practice, such rigid grouping tends to cluster near dense objects and far sparse objects from the same viewpoint. Moreover, this method is designed for single-category tasks and requires further exploration in multi-category scenarios.

For the issue of ambiguous foreground-background decision boundaries, Li et al. (2023); Zhang et al. (2024) attempt to add high-confidence pseudo labels to unreliable background regions. However, pseudo labels themselves contain noise. Additionally, due to the high-confidence filtering mechanism, pseudo labels often focus on simple patterns and lack diversity, limiting model generalization. These methods also fail to address the noisy foreground issue.

Based on these observations, we propose a novel Adaptive Multi-Prototype Grouping Alignment framework for multi-category 3D UDA to address the two issues mentioned above. For the insufficient cross-domain feature alignment issue, we design a data-driven grouping model for each category that automatically identifies feature groups with different representations. They update dynamically to adapt to domain differences. During training, we extract features from both domains and assign them to the corresponding cross-domain learnable prototypes through the grouping model. We use contrastive learning between features and learnable prototypes to reduce cross-domain feature distance within groups while expanding inter-group differences. Moreover, we design two important components to address the issue of ambiguous foreground-background decision boundaries: (1) We propose the Noise Background Hybrid Augmentation strategy to alleviate the noisy background issue by introducing source labeled instances with high-confidence pseudo labels to background regions potentially containing positive samples. The introduction of source instances mitigate the noise in pseudo labels while enhance their diversity. (2) We propose a Noise Foreground Contrastive Learning strategy that leverages prototype guidance to push low-confidence ambiguous features away from decision boundaries, effectively mitigating the noisy foreground issue.

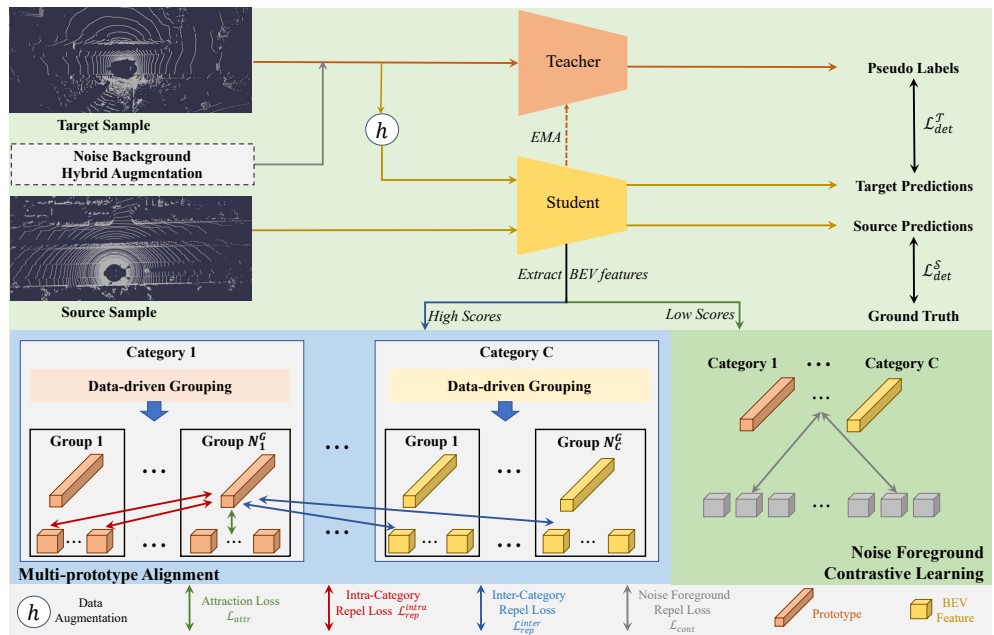

Figure 2: Overview of the proposed method. During training, we first group BEV features using our Data-driven Grouping method, and then align these features with corresponding category prototypes through Multi-prototype Alignment. Subsequently, we further optimize feature representations with Noise Foreground Contrastive Learning, and also enhancing the quality of pseudo labels using Noise Background Hybrid Augmentation. Additionally, both the grouping model and target domain pseudo labels are updated every $K$ epochs to adapt to evolving feature distributions.

The main contributions of this paper include:

- We propose a novel Adaptive Multi-Prototype Grouping Alignment framework for multi-category 3D UDA that automatically identifies feature groups with different representations for effective cross-domain alignment.

- We design two components to address ambiguous foreground-background decision boundaries: a Noise Background Hybrid Augmentation strategy that integrates source labeled instances to enhance pseudo label quality, and a Noise Foreground Contrastive Learning strategy that pushes low-confidence features away from decision boundaries using prototype guidance.

- Extensive experiments on 3D UDA scenarios across multiple datasets validate the effectiveness of our method.

## 2 RELATED WORK

**LiDAR-based 3D Object Detection.** LiDAR-based 3D object detection has been widely studied in recent years, which aims to detect objects in 3D space using point clouds. The existing methods can be roughly grouped into two categories: point-based methods and voxel-based methods. Point-based methods Shi et al. (2019) leverage the rich information in original point clouds and typically use PointNet series Qi et al. (2017a;b) to directly extract point-wise features. On the other hand, voxel-based methods Yan et al. (2018); Lang et al. (2019); Zhou & Tuzel (2018) divide point clouds into regular grids, encode features using sparse convolution, and map them to bird's eye view space. These methods usually achieve faster processing speeds. Some methods also combine the advantages of both points and voxels Shi et al. (2020; 2023). However, these excellent methods often experience sharp performance drops when applied to target domains Wang et al. (2020). Since current state-of-the-art methods are primarily voxel-based, we adopt SECOND-IOU Yan et al. (2018); Yang et al. (2021) and PVRCNN Shi et al. (2020) as our base detector for experiments.

**Domain Adaptive 3D Object Detection.** In recent years, UDA has been widely explored in 2D Ganin & Lempitsky (2015); Zhuo et al. (2022); Zhi et al. (2024); Deng et al. (2023), mainly using adversarial training and statistical methods Creswell et al. (2018); Long et al. (2017). Recently, several methods have also been proposed for 3D UDA. SN Wang et al. (2020) reduces size deviation by adjusting source domain data based on target domain size information. Since target domain size information is often hard to obtain, PLS Chen et al. (2024a) uses pseudo label information to guide source domain data scaling, while Sailor Malić et al. (2023) adapts anchor boxes to target domain size by searching for optimal features. LD Wei et al. (2022) uses point cloud distillation to adapt models from high-density to low-density point clouds. DTS Hu et al. (2023) proposes a teacher-student framework with point cloud beam random resampling to address differences in point cloud density. The ST3D series Yang et al. (2021; 2022) introduces a self-training pipeline that updates pseudo labels using a memory bank. MLC-net Luo et al. (2021) presents a teacher-student framework for multi-level consistency alignment. 3D-CoCo Yihan et al. (2021) employs cross-domain contrastive learning for transferable representations. GPA Li et al. (2023) groups features by observation angle for domain alignment. ReDB Chen et al. (2023) extends 3D UDA to multi-category settings, addressing unreliable pseudo-labels and class imbalance through cross-domain examination and class-balanced self-training. PERE Zhang et al. (2024) tackles unreliable multi-category pseudo labels by replacing them and generating additional proposals, using cross-domain triplet loss for category feature alignment. Despite these advances, most methods focus on single-category domain adaptation, neglecting multi-category adaptation challenges. Additionally, existing approaches don't fully address adaptive alignment in feature space, while our method achieves comprehensive feature alignment through a data-driven grouping approach.

## 3 METHODOLOGY

### 3.1 PROBLEM STATEMENT

Domain Adaptive 3D Object Detection aims to adapt a 3D detector pretrained on a labeled source domain $D_S = \{X_i^S, Y_i^S\}_{i=1}^{N_S}$ to an unlabeled and target domain $D_T = \{X_i^T\}_{i=1}^{N_T}$, where $X_i^S$ and $X_i^T$ are the input point clouds, $Y_i^S$ are corresponding labels, and $N_S$ and $N_T$ are the number of samples in each domain, respectively. The 3D box label is represented as $(x, y, z, l, w, h, \theta, c)$, where $x, y, z$ represents the box center, $l, w, h$ represents the box size, $\theta$ represents the orientation, and $c$ represents the category.

### 3.2 OVERALL FRAMEWORK

Our method starts by pretraining a 3D detector on the source domain $D_S$ with two domain adaptation data augmentation methods: ROS Yang et al. (2021) and RBRS Hu et al. (2023). For stable training, we adopt a teacher-student framework Hu et al. (2023) as our base structure. This framework consists of a non-trainable teacher model and a trainable student model with identical network architecture, both initialized with the source pretrained model. The weights of the teacher model are updated using exponential moving average (EMA). Since mainstream 3D detectors Yan et al. (2018); Shi et al. (2020) typically project sparse 3D features to bird's eye view (BEV), and previous works Yihan et al. (2021); Li et al. (2023); Hu et al. (2023) have shown good transferability of BEV features, we sample BEV features from the Region of Interest (RoI) head for feature alignment.

As shown in Fig. 2, during training, we first group the BEV features with our Data-driven Grouping method. We then align features and prototypes using our Multi-prototype Alignment method. We further optimize them through Noise Foreground Contrastive Learning. We generate target domain pseudo labels using the teacher model every $K$ epochs and update the data-driven grouping model accordingly. We also apply Noise Background Hybrid Augmentation to process the pseudo labels. The overall prototype loss $\mathcal{L}_{proto}$ is defined as:

$$\mathcal{L}_{proto} = \beta_1 \mathcal{L}_{attr} + \mathcal{L}_{rep} + \beta_2 \mathcal{L}_{cont}, \qquad (1)$$

where $\mathcal{L}_{attr}$ and $\mathcal{L}_{rep}$ are the attraction and repulsion losses in feature alignment, and $\mathcal{L}_{cont}$ is the noise foreground contrastive loss. $\beta_1$ and $\beta_2$ are the balance coefficients.

To reduce error accumulation, we also train the student model with the labeled source domain:

$$\mathcal{L}_{det}^S = \mathcal{L}_{reg}^S + \mathcal{L}_{cls}^S, \qquad (2)$$

where $\mathcal{L}_{reg}^S$ and $\mathcal{L}_{cls}^S$ are the regression and classification losses in the source domain, respectively.

The overall training loss is:

$$\mathcal{L} = \mathcal{L}_{det}^S + \mathcal{L}_{det}^T + \lambda\mathcal{L}_{proto}, \tag{3}$$

where $\mathcal{L}_{det}^T$ is the detection loss in the target domain with pseudo labels, and $\lambda$ is the weight of the prototype loss.

### 3.3 DATA-DRIVEN GROUPING

Although multi-prototypes can better represent complex category features, grouping these category features remains challenging. Inspired by the promising performance of probability distributions in unsupervised learning Shin et al. (2024); Malić et al. (2023), we use a data-driven method to adaptively group features. We implement a Gaussian Mixture Model (GMM) as the grouping model, which is dynamically updated during training. We set up GMM models with $\{N_c^G\}_{c=1}^C$ Gaussian distributions, where $C$ is the number of categories. The number of Gaussian distributions in each GMM model equals the number of learnable prototypes for each category, enabling one-to-one correspondence.

**Adaptive Group.** During training, we extract BEV features of the point cloud and use GMM to group them. For the $k$-th BEV feature $f_k^c$, we find the corresponding GMM model based on its predicted category $c$, and then calculate the probability that it belongs to the $i$-th Gaussian distribution, which is also the probability that it belongs to the $i$-th prototype of that category, defined as:

$$p_i^k = \frac{\mathcal{N}(f_k^c; \mu_i, \Sigma_i)}{\sum_j^{N_c^G} \mathcal{N}(f_k^c; \mu_j, \Sigma_j)}, \tag{4}$$

where $N_c^G$ is the number of Gaussian distributions in the GMM model for category $c$, and $\mathcal{N}(f_k^c; \mu_j, \Sigma_j)$ is the Gaussian distribution defined as:

$$\mathcal{N}(f_k^c; \mu_j, \Sigma_j) = \frac{exp(-\frac{1}{2}(f_k^c - \mu_j)^T \Sigma_j^{-1}(f_k^c - \mu_j))}{\sqrt{(2\pi)^{N_c^G}|\Sigma_j|}}, \tag{5}$$

where $\mu_j$ and $\Sigma_j$ are the mean and covariance of the $j$-th Gaussian distribution, respectively. The group label of feature $f_k^c$ is defined as:

$$g_c^k = \underset{i \in \{1,2,...,N_c^G\}}{\arg\max} p_i^k . \tag{6}$$

This approach allows us to adaptively assign a group label to each BEV feature.

**Dynamic Update.** Considering that the training data fluctuates significantly in each batch, updating the GMM every batch may affect its performance. To prevent the feature distribution from being dominated by source domain features, we select equal amounts of source and target domain data to fit the GMM model parameters $\{\mu_i, \Sigma_i, \pi_i\}_{i=1}^{N_c^G}$ for each category. Both initialization and updates use the Expectation-Maximization (EM) algorithm Dempster et al. (1977). For the update stage, we use the current epoch of BEV features and the GMM model to perform the EM algorithm to obtain the updated GMM model parameters for each category, $\{\hat{\mu}_i, \hat{\Sigma}_i, \hat{\pi}_i\}_{i=1}^{N_c^G}$.

### 3.4 MULTI-PROTOTYPE ALIGNMENT

At the beginning of training, we randomly initialize a set of learnable prototypes for each category, $\{\{\mathcal{G}_i^c\}_{i=1}^{N_c^G}\}$, with the number matching the grouping model. We assign BEV features from the source and target domains to the corresponding group , and then perform intra-group attraction and inter-group repulsion between the features and prototypes.

We push features towards the corresponding prototypes to encourage similar features to cluster together. The attraction loss Li et al. (2023); Shin et al. (2024) between intra-group features and prototypes is defined as:

$$\mathcal{L}_{attr} = \frac{1}{C}\sum_{c=1}^C \frac{1}{|F_c|} \sum_{f_k^c \in F_c} \sum_{i=1}^{N_c^G} \mathbb{I}(g_c^k = i) \cdot (1 - \cos(f_k^c, \mathcal{G}_i^c)), \tag{7}$$

where $\mathbb{I}(\cdot)$ is the indicator function, which is 1 if the condition is true and 0 otherwise. $|F_c|$ is the number of features in category $c$, and $f_k^c$ is the $k$-th feature in category $c$.

We push features away from the prototypes of other groups to enhance discriminative ability. We first separate different prototypes within the same group to ensure that each group learns distinct features. The intra-category repulsion loss Li et al. (2023) is defined as:

$$\mathcal{L}_{rep}^{intra} = \frac{1}{C} \sum_{c=1}^{C} \frac{1}{|F_c|} \sum_{f_k^c \in F_c} \sum_{i=1}^{N_c^G} \mathbb{I}(g_c^k \neq i) \cdot \max(0, \cos(f_k^c, \mathcal{G}_i^c)). \tag{8}$$

For features and prototypes of different categories, we apply repulsion to strengthen inter-category discriminative ability. The inter-category repulsion loss is defined as:

$$\mathcal{L}_{rep}^{inter} = \frac{1}{C} \sum_{c=1}^{C} \frac{1}{|F_c|} \sum_{f_k^c \in F_c} \sum_{c' \neq c} \sum_{i=1}^{N_{c'}^G} \max(0, \cos(f_k^c, \mathcal{G}_i^{c'})). \tag{9}$$

The overall repulsion loss is:

$$\mathcal{L}_{rep} = \beta_3 \mathcal{L}_{rep}^{intra} + \beta_4 \mathcal{L}_{rep}^{inter}, \tag{10}$$

where $\beta_3$ and $\beta_4$ are the balance coefficients.

### 3.5 NOISE FOREGROUND CONTRASTIVE LEARNING

During training, the model often misclassifies confusing background regions as low-confidence foreground, limiting its ability to distinguish between foreground and background. Since background features are typically dispersed and complex, making it difficult to construct effective background prototypes, we instead repel low-confidence foreground features from existing prototypes to enhance the discriminative ability. Our experiments (Table 5) show that repelling low-confidence foreground features is more effective than constructing dedicated background prototypes.

We define foreground features with confidence below $T_{neg}$ as low-confidence foreground features and repel them from all category prototypes:

$$\mathcal{L}_{cont} = \frac{1}{|F_{low}|} \sum_{f_k^c \in F_{low}} \sum_{c=1}^{C} \sum_{i=1}^{N_c^G} \max(0, \cos(f_k^c, \mathcal{G}_i^c)), \tag{11}$$

where $F_{low}$ represents the set of low-confidence foreground features. This loss encourages the model to push ambiguous low-confidence foreground features away from category prototypes, thereby improving the robustness and foreground-background separation.

### 3.6 NOISE BACKGROUND HYBRID AUGMENTATION

Self-training methods typically select high-confidence predictions as pseudo labels for training to reduce noise influence. However, they inevitably miss some low-confidence positive samples, wrongly treating them as background. We propose the Noise Background Hybrid Augmentation module to mitigate this negative supervision effect.

Specifically, we extract instances from the source domain and high-confidence instances from the target domain to build a hybrid instance bank. We identify pseudo labels with confidence scores in the range $[T_{neg}, T_{pos}]$ as noise background, where $T_{neg}$ and $T_{pos}$ are pre-defined thresholds for selecting negative and positive samples. During training, we randomly select samples from the bank and paste them into noise background regions of the current target frame. Following Li et al. (2023); Zhang et al. (2024), we ensure selected samples match the original region through scaling and rotation operations. We use hyperparameter $\gamma$ to control the probability of selecting source domain samples with labels. By enriching the target domain with diverse and reliable foreground instances, our method effectively stabilizes the self-training process.

Table 1: Cross-dataset 3D object detection results with SECOND-IoU backbone for Waymo → KITTI and nuScenes → KITTI scenarios across different difficulty levels. For fair comparison, all methods use the same pre-trained model. Methods marked with * are originally designed for single-category detection and we modified for multi-category detection. ReDB[†] follows the original 120-epoch training strategy from its original paper, while all other methods use 30 epochs as in their original papers. PERE[‡] is our implementation based on the original paper. The best results are shown in **bold** and the second best are underlined.

| Task | Method | Easy | | Moderate | | Hard | |
|------|--------|------|--|----------|--|------|--|
| | | $AP_{BEV}/AP_{3D}$ | Closed Gap | $AP_{BEV}/AP_{3D}$ | Closed Gap | $AP_{BEV}/AP_{3D}$ | Closed Gap |
| W → K | Source Only | 46.88/28.81 | -/- | 41.02/25.03 | -/- | 38.71/23.60 | -/- |
| | ST3D* | 65.34/48.44 | 64.0%/46.6% | 58.38/43.66 | 70.7%/55.2% | 55.75/41.68 | 73.6%/58.9% |
| | ST3D++* | 66.20/53.71 | 66.9%/59.1% | 59.37/47.52 | 74.8%/66.5% | 56.92/45.36 | 78.5%/70.5% |
| | DTS* | 65.68/55.64 | 65.1%/63.5% | 59.85/49.43 | 76.7%/72.2% | 57.08/47.06 | 79.2%/76.6% |
| | ReDB[†] | 63.41/55.46 | 57.3%/63.3% | 56.06/47.13 | 61.4%/65.1% | 54.11/44.93 | 66.3%/69.4% |
| | PERE[‡] | 66.35/54.26 | 67.5%/60.4% | 58.65/47.92 | 71.8%/67.6% | 56.17/45.42 | 75.2%/71.0% |
| | Ours | **68.33/58.90** | **74.4%/71.4%** | **61.33/51.46** | **82.7%/78.1%** | **58.72/48.66** | **86.2%/81.6%** |
| | Oracle | 75.79/70.96 | -/- | 65.57/58.88 | -/- | 61.93/54.27 | -/- |
| N → K | Source Only | 21.16/9.22 | -/- | 17.41/7.43 | -/- | 17.51/7.12 | -/- |
| | ST3D* | 42.65/36.09 | 39.3%/43.5% | 37.53/30.03 | 41.9%/43.9% | 36.51/28.05 | 42.8%/44.3% |
| | ST3D++* | 50.58/41.54 | 53.8%/52.4% | 43.35/34.30 | 53.8%/52.2% | 42.03/32.15 | 55.2%/53.1% |
| | DTS* | 53.72/37.78 | 59.6%/46.3% | 46.26/30.95 | 59.8%/45.7% | 44.27/29.17 | 60.2%/46.7% |
| | ReDB[†] | 46.94/38.42 | 47.1%/47.3% | 40.62/31.17 | 48.1%/46.1% | 39.18/29.83 | 48.9%/48.1% |
| | PERE[‡] | 51.19/42.32 | 55.0%/53.7% | 44.32/35.27 | 55.9%/54.0% | 42.47/33.56 | 56.2%/56.1% |
| | Ours | **56.91/48.33** | **65.4%/63.4%** | **48.06/39.43** | **63.6%/62.1%** | **46.08/37.33** | **64.3%/64.0%** |
| | Oracle | 75.79/70.96 | -/- | 65.57/58.88 | -/- | 61.93/54.27 | -/- |

Table 2: Cross-dataset 3D object detection results with SECOND-IoU backbone for Waymo → nuScenes scenario.

| Method | $AP_{BEV}$ | Closed Gap | $AP_{3D}$ | Closed Gap |
|--------|-----------|-----------|----------|-----------|
| Source Only | 11.41 | – | 7.30 | – |
| ST3D* | 17.35 | 31.2% | 10.68 | 23.8% |
| ST3D++* | 17.44 | 32.0% | 11.07 | 26.5% |
| DTS* | 17.46 | 32.1% | 10.60 | 23.2% |
| ReDB[†] | 14.67 | 17.1% | 8.20 | 6.3% |
| PERE[‡] | 17.42 | 31.8% | 10.30 | 21.1% |
| Ours | **17.73** | **33.0%** | **11.21** | **27.5%** |
| Oracle | 30.45 | – | 21.54 | – |

Table 3: Performance comparison of various methods on the Waymo → KITTI adaptation scenario with PVRCNN. We report the AP at **Moderate** difficulty.

| Method | $AP_{3D}$ | $AP_{BEV}$ |
|--------|----------|-----------|
| ST3D* | 53.29 | 63.33 |
| ST3D++* | 53.17 | 63.99 |
| DTS* | 52.68 | 63.42 |
| PERE[‡] | 50.73 | 58.33 |
| Ours | **54.49** | **64.65** |

# 4 EXPERIMENTS

## 4.1 EXPREIMENTAL SETTINGS

**Datasets.** We conduct experiments on three popular autonomous driving datasets: KITTI Geiger et al. (2012), Waymo Sun et al. (2020), and nuScenes Caesar et al. (2020). Following previous works Yang et al. (2021; 2022); Hu et al. (2023); Li et al. (2023); Shin et al. (2024); Zhang et al. (2024), we conduct experiments in three domain adaptation scenarios (Waymo → KITTI, nuScenes → KITTI, Waymo → nuScenes) and evaluate all models on the validation set of each target dataset. Please refer to the supplements for more datasets details.

**Evaluation Metrics.** Following Yang et al. (2021); Chen et al. (2023); Zhang et al. (2024), we use the official KITTI evaluation metric for 3D object detection over 40 recall positions with IoU thresholds of 0.7, 0.5 and 0.5 for Car, Pedestrian, and Cyclist. In the nuScenes dataset, we merge the bicycle and motorcycle categories into the Cyclist category for consistency across datasets.

We also adopt the Closed Gap Yang et al. (2021) metric, which is defined as $Closed\ Gap = \frac{AP_{model} - AP_{source}}{AP_{oracle} - AP_{source}} \times 100\%$.

**Implementation Details.** We use OpenPCDet Team (2020) as our codebase to pre-train detectors on the source domain. In the self-training process, we fine-tune the model for 30 epochs. We use Adam optimizer Kingma & Ba (2014) with one cycle scheduler to adjust the learning rate, which was set to 0.003 for pretraining and 0.0015 for fine-tuning. Please refer to the supplements for more implementation details.

**Compared Methods.** We compare our proposed method with existing state-of-the-art methods Yang et al. (2021; 2022); Hu et al. (2023); Chen et al. (2023); Zhang et al. (2024). In addition, we also report the results of **Source Only** and **Oracle**. **Source Only** indicates directly evaluating the source domain pre-trained model on the target domain, while **Oracle** is the result of fully supervised training on the target domain, serving as an upper bound for performance.

## 4.2 COMPARISON WITH STATE-OF-THE-ART METHODS

**Main Results.** As shown in Table 1, our method demonstrates exceptional performance across all challenging domain adaptation scenarios, consistently achieving the best results in both $AP_{BEV}$ and $AP_{3D}$ metrics across all difficulty levels. Particularly in the nuScenes $\rightarrow$ KITTI scenario, our method achieves average improvements of 2.27% in $AP_{BEV}$ over the second-best method DTS Hu et al. (2023) and 4.65% in $AP_{3D}$ over the second-best method PERE Zhang et al. (2024) respectively. As shown in Table 2, even in the challenging Waymo $\rightarrow$ nuScenes scenario with significant LiDAR beam density differences, our method consistently maintains the best performance.

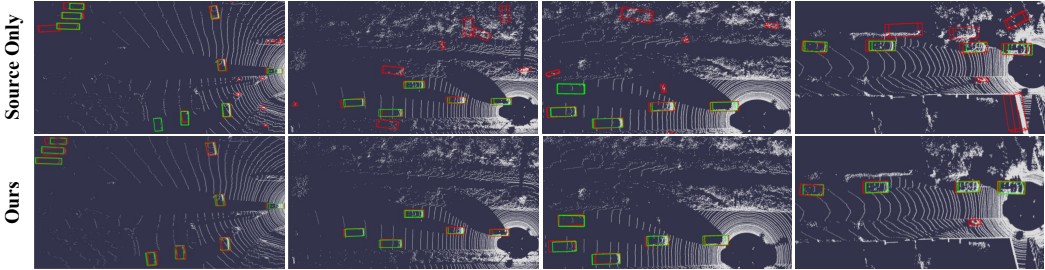

Figure 3: Qualitative results on nuScenes $\rightarrow$ KITTI. The predictions and ground truths are painted in red and green, respectively.

**Evaluation Across Different Architectures.** Table 3 verifies the effectiveness of our method with a different backbone architecture. Using PVRCNN Shi et al. (2020) for Waymo $\rightarrow$ KITTI adaptation, our method achieves the best average $AP_{BEV}$ and $AP_{3D}$, improving upon the second-best method (ST3D Yang et al. (2021), ST3D++ Yang et al. (2022)) by 0.66% and 1.20% respectively. This demonstrates the general applicability of our method across different model architectures.

## 4.3 ABLATION STUDIES

In this section, all ablation experiments are based on SECOND-IoU, conducted on nuScenes $\rightarrow$ KITTI, and evaluated for average results under moderate difficulty.

**Main Ablation Studies.** Table 4 shows the contribution of each component in our method. Starting with the baseline (a), adding Noisy Background Hybrid Augmentation (NBHA) in setting (b) improves $AP_{3D}$ by 2.78% and $AP_{BEV}$ by 4.20% by reducing noisy supervision from noisy background regions. The attraction loss in setting (c) brings additional gains of 0.47% in $AP_{3D}$ and 0.88% in $AP_{BEV}$. The repulsion loss in setting (d) further improves results by 0.65% in $AP_{3D}$ and 0.89% in $AP_{BEV}$. Setting (e) confirms that attraction loss and repulsion work synergistically. Setting (f) shows that without NBHA, performance drops substantially even with all other components present, highlighting NBHA's importance in addressing noise backgrounds. Our complete method (g) with Noisy Foreground Contrastive Learning (NFCL) achieves the best results, with further improvements of 1.45% in $AP_{3D}$ and 0.96% in $AP_{BEV}$.

Table 4: Ablation study of different components in our method.

| Setting | NBHA | Attr | Repel | NFCL | $AP_{3D}$ | $AP_{BEV}$ |
|---------|------|------|-------|------|-----------|------------|
| (a) |  |  |  |  | 34.08 | 41.13 |
| (b) | ✓ |  |  |  | 36.86 | 45.33 |
| (c) | ✓ | ✓ |  |  | 37.33 | 46.21 |
| (d) | ✓ |  | ✓ |  | 37.31 | 45.39 |
| (e) | ✓ | ✓ | ✓ |  | 37.98 | 47.10 |
| (f) |  | ✓ | ✓ | ✓ | 35.23 | 42.42 |
| (g) | ✓ | ✓ | ✓ | ✓ | **39.43** | **48.06** |

Table 5: Ablation study on the impact of NBHA. CA represents the complementary augmentation.

| Method | $AP_{BEV}$ | $AP_{3D}$ |
|--------|------------|-----------|
| Baseline | 41.13 | 34.08 |
| CA | 46.03 | 37.31 |
| Ours | **48.06** | **39.43** |

Table 6: Ablation study of different prototype strategies.

| Setting | Proto | Multi | K-Means | GMM | $AP_{3D}$ | $AP_{BEV}$ |
|---------|-------|-------|---------|-----|-----------|------------|
| (a) |  |  |  |  | 34.08 | 41.13 |
| (b) | ✓ |  |  |  | 36.03 | 47.41 |
| (c) |  |  |  | ✓ | 38.11 | 47.25 |
| (d) | ✓ | ✓ | ✓ |  | 38.25 | 45.94 |
| (e) | ✓ | ✓ |  | ✓ | **39.43** | **48.06** |

Table 7: Ablation study on the impact of NFCL.

| Method | $AP_{BEV}$ | $AP_{3D}$ |
|--------|------------|-----------|
| Baseline | 41.13 | 34.08 |
| BG Proto | 47.26 | 37.02 |
| Ours | **48.06** | **39.43** |

**Analysis of Prototype Alignment Architecture.** Table 6 compares different prototype strategies. Setting (b) shows that using only single prototype for alignment achieves limited improvement, indicating that single prototype struggles to represent complex category features effectively. Setting (c), which uses only GMM distribution for alignment without explicit prototypes, performs better than single prototype alignment with 38.11% $AP_{3D}$, demonstrating the effectiveness of modeling feature distribution. Setting (d) combines multiple learnable prototypes with K-Means clustering, yielding comparable $AP_{3D}$. Our proposed method in setting (e), which integrates GMM-based grouping with multiple learnable prototypes, achieves the best performance. This demonstrates that GMM provides more effective feature grouping than K-Means, and when combined with multiple learnable prototypes, enables more precise alignment of complex feature distributions across domains.

**Analysis of Noise Background Hybrid Augmentation.** Table 5 compares our NBHA with other method. Complementary Augmentation (CA) Zhang et al. (2024) only uses high-confidence instances from the target domain to fill noisy backgrounds. In contrast, our NBHA incorporates instances from the source domain in addition to target domain instances, effectively alleviating the noise by introducing more reliable supervision signals. The results demonstrate that NBHA achieves superior performance with improvements of 2.03% in $AP_{BEV}$ and 2.12% in $AP_{3D}$ compared with CA.

**Analysis of Noise Foreground Contrastive Learning.** Table 7 shows the effectiveness of our NFCL approach. Background features vary greatly across different scenes, making them difficult to represent with prototypes. While using background prototypes (BG Proto) improves over the baseline, it cannot accurately capture background distributions. Our NFCL uses noisy foreground features to optimize prototypes through contrastive learning, improving the model's ability to separate foreground from background. NFCL achieves better performance with 2.41% improvement in $AP_{3D}$ and 0.80% in $AP_{BEV}$ compared with BG Proto.

**Qualitative Results.** Fig. 3 presents qualitative results. Our method generates clean and more accurate predictions, demonstrating the effectiveness of our method.

## 5 CONCLUSION

In this paper, we propose a novel Adaptive Multi-Prototype Grouping Alignment framework for unsupervised domain adaptation on multi-category 3D detection. Our method addresses the challenges of inadequate cross-domain feature alignment and ambiguous foreground-background decision boundaries. We design a data-driven grouping model that dynamically updates feature groups and assigns learnable prototypes to each group. Additionally, we introduce Noise Background Hybrid Augmentation and Noise Foreground Contrastive Learning to enhance feature alignment and mitigate the impact of unreliable pseudo-labels. Extensive experiments validate the effectiveness of our method. In future work, we plan to explore the application on multi-modal 3D UDA tasks.

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

## A    APPENDIX

In this supplementary material, we provides more details of our experiments and additional experimental results and analysis.

## B    USE OF LARGE LANGUAGE MODELS (LLMS)

We used a large language model (e.g., DeepSeek) solely as a general-purpose writing assistant for minor language polishing (grammar, phrasing, and formatting consistency). The LLM did not contribute to research ideation, problem formulation, methodological design, data analysis, result interpretation, or the drawing of scientific conclusions. All scientific content, including ideas, methods, experiments, and claims, originates from the authors. The authors verified and edited all text and take full responsibility for the content.

## C    DATASETS

We conduct experiments on three widely used autonomous driving datasets: KITTI Geiger et al. (2012), Waymo Sun et al. (2020), and nuScenes Caesar et al. (2020). These datasets were collected using different LiDAR configurations across various geographical locations. The KITTI dataset was collected in Germany using a 64-beam Velodyne LiDAR and contains 7,481 labeled frames for training and 7,518 unlabeled frames for testing. Following Yang et al. (2021; 2022), we divide the training set into two disjoint subsets: 3,712 samples for training and 3,769 samples for validation. The Waymo dataset was collected in the United States using one 64-beam and four 200-beam LiDAR sensors, which consists of 798 sequences (over 150,000 frames) for training and 202 sequences (approximately 40,000 frames) for validation. Following Yang et al. (2021; 2022), we randomly subsample 20% of the original training set. The nuScenes dataset, collected in both the United States and Singapore using a 32-beam LiDAR, contains 28,130 samples for training and 6,019 samples for validation.

## D    MORE IMPLEMENTATION DETAILS

We use OpenPCDet Team (2020) as our codebase to pre-train detectors on the source domain. In the pre-training process, the learning rate is set to $1.0 \times 10^{-2}$. In the self-training process, we fine-tune the model for 30 epochs. We use Adam optimizer Kingma & Ba (2014) with one cycle scheduler to adjust the learning rate, which was set to 0.003 for pretraining and 0.0015 for fine-tuning. we use Adam optimizer Kingma & Ba (2014) with one cycle scheduler for 30 epochs. The learning rate is set to $1.5 \times 10^{-3}$. We update pseudo labels and GMM models every $K = 2$ epochs. The threshold of pseudo labels $[T_{neg}, T_{pos}]$ are consistent with previous works Yang et al. (2021; 2022); Hu et al. (2023), between 0.2-0.5. While in the Waymo $\rightarrow$ nuScenes task, we do not update pseudo labels as previous works Yang et al. (2021; 2022) for fair comparisons. We empirically set the balance coefficients $\lambda$, $\beta_1$, $\beta_2$, $\beta_3$ ,$\beta_4$ and $\gamma$ to be 0.1, 1.0, 0.5, 0.5 ,1.5 and 0.7, respectively.

## E    ADDITIONAL EXPERIMENTAL RESULTS

**Sensitivity Analysis of hyperparameters.** We conduct a sensitivity analysis on six key hyperparameters in our method: $\lambda$, $\beta_1$, $\beta_2$, $\beta_3$, $\beta_4$, and $\gamma$. The values used in our final model are 0.1, 1.0, 0.5, 0.5, 1.5, and 0.7, respectively, under which we achieve a 3D AP of 39.43. To evaluate the robustness of our method to hyperparameter variations, we individually vary each parameter while keeping the others fixed and report the resulting 3D AP in Table 8. The results show that our method maintains stable performance under different settings, indicating good robustness. Notably, $\gamma$ and $\beta_2$ exhibit more significant influence: increasing $\gamma$ from 0.5 to 0.7 (ours) leads to a substantial gain of 2.69 points in 3D AP, and decreasing $\beta_2$ from 0.5 (ours) to 0.1 brings moderate improvement. These observations suggest that careful tuning of these parameters can yield further performance gains.

**Analysis of Results Under Different Training Settings.** Fair comparison between different methods is non-trivial due to inconsistencies in codebases and pretraining configurations. Specifically,

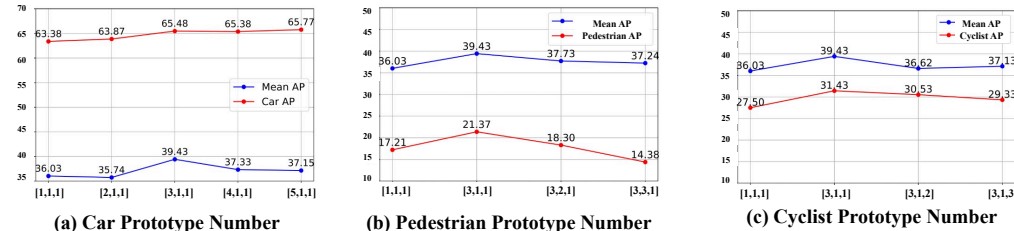

Figure 4: Class AP and mean AP (%) under varying prototype numbers for each class. (a) Increasing Car prototypes improves performance up to 3, then saturates. (b, c) Pedestrian and Cyclist perform best with a single prototype; more prototypes degrade performance due to redundancy.

Table 8: Sensitivity analysis of hyperparameters. The performance is reported in terms of 3D AP (%) under different values. The values used in our final model are highlighted in bold.

| Hyperparameter | Value | 3D AP (%) |
|---|---|---|
| $\lambda$ | 0.05 | 37.14 |
| | 0.5 | 37.58 |
| | **0.1 (ours)** | **39.43** |
| $\beta_1$ | 0.5 | 37.35 |
| | 1.5 | 36.87 |
| | **1.0 (ours)** | **39.43** |
| $\beta_2$ | 0.1 | 38.40 |
| | 1.0 | 38.27 |
| | **0.5 (ours)** | **39.43** |
| $\beta_3$ | 0.1 | 37.53 |
| | 1.0 | 37.43 |
| | **0.5 (ours)** | **39.43** |
| $\beta_4$ | 1.0 | 37.71 |
| | 2.0 | 38.10 |
| | **1.5 (ours)** | **39.43** |
| $\gamma$ | 0.5 | 36.74 |
| | 1.0 | 38.01 |
| | **0.7 (ours)** | **39.43** |

DTS Hu et al. (2023) is implemented with `spconv 2.x` Contributors (2022) and adopts RBRS Hu et al. (2023) for pretraining, whereas ST3D Yang et al. (2021) is based on `spconv 1.x` and does not incorporate RBRS. Such differences introduce performance gaps unrelated to algorithmic design, making direct comparison less informative.

To ensure fairness, we re-implement and rerun all methods under a unified codebase and consistent pretraining settings. As shown in Table 9, the impact of both codebase version and RBRS is substantial. For instance, when re-running ST3D with `spconv 2.x`, its 3D AP drops from 23.27 to 20.28, indicating that the codebase upgrade alone can negatively affect performance. However, introducing RBRS consistently improves performance across both codebases. For ST3D, RBRS boosts the 3D AP from 23.27 to 28.67 under `spconv 1.x`, and from 20.28 to 30.03 under `spconv 2.x`. A similar trend is observed for ST3D++, where RBRS improves the AP from 28.40 to 33.20 under `spconv 1.x`, and from 30.94 to 34.30 under `spconv 2.x`.

These results demonstrate that both the spconv version and the use of RBRS significantly affect model performance. Importantly, our final setting—`spconv 2.x` with RBRS—not only enables a fair and modern implementation, but also achieves the best 3D AP among all combinations for both ST3D and ST3D++, validating the effectiveness and robustness of our approach.

Table 9: 3D AP (%) comparison of ST3D and ST3D++ under different combinations of `spconv` versions and RBRS usage. Abbreviations: v1 = `spconv 1.x`, v2 = `spconv 2.x`, R = with RBRS, Ours = v2 + RBRS.

| Method | v1 (Orig.) | v1+R | v2 | v2+R (Ours) |
|---|---|---|---|---|
| ST3D | 23.27 | 28.67 | 20.28 | **30.03** |
| ST3D++ | 28.40 | 33.20 | 30.94 | **34.30** |

**Analysis of Prototype Numbers.** We consistently observe that the group configuration [3,1,1] yields the best overall performance across all evaluated settings. This can be attributed to the varying levels of complexity and domain discrepancy among different object categories. As shown in Figure 4(a), increasing the number of prototypes for the Car class from 1 to 3 significantly improves both Car AP and mean AP. This suggests that a richer prototype representation is beneficial for modeling the higher intra-class variation and larger domain gap observed in the Car class. However, allocating more than three prototypes to the Car class introduces imbalance, which slightly degrades the mean AP.

In contrast, Figures 4(b) and (c) show that the Pedestrian and Cyclist classes perform best when assigned a single prototype. Increasing the number of prototypes for these relatively simpler classes leads to performance degradation, likely due to redundancy and overfitting. For example, the [3,3,1] and [3,1,3] configurations consistently underperform compared to [3,1,1]. This confirms that allocating excessive prototypes to low-complexity classes not only impairs their own AP but also negatively affects the overall detection performance. These findings underscore the importance of class-specific prototype allocation, where more complex classes benefit from multiple prototypes, while simpler classes are better represented with fewer.

