# OpenReview forum: "Adaptive Multi-Prototype Grouping Alignment for Domain Adaptive 3D Detection"
_ICLR.cc/2026/Conference — ICLR 2026 Conference Withdrawn Submission_

### Official Review · Reviewer_21dT · 2025-10-31

**Soundness:** 4
**Presentation:** 3
**Contribution:** 2
**Rating:** 4
**Confidence:** 3

**Summary:**

This paper presents a new framework for unsupervised domain adaptation (UDA) in 3D object detection.

The proposed method, Adaptive Multi-Prototype Grouping Alignment, aims to address insufficient feature alignment and ambiguous foreground-background boundaries. It introduces a GMM-based approach to dynamically group features and align them with multiple learnable prototypes per category, a hybrid augmentation technique to improve pseudo-label quality, and a contrastive loss to refine the decision boundary.

**Strengths:**

1. Clear motivation. Reasonable solution.

2. Comprehensive experiments.

**Weaknesses:**

1. Novelty vs. GPA-3D, 3D-CoCo, and PERE is somewhat incremental: the core contribution is largely an improved engineering of prototype alignment with GMM grouping and augmentation, without any deeper theoretical insight into when multi-prototype grouping is necessary or optimal.

2. The method introduces a considerable number of hyperparameters. The appendix experiments show that performance is somewhat sensitive to these.

3. No discussion over computational overhead (memory, FLOPS, FPS, etc.)

**Questions:**

it would be nice if the authors can provide responses over the mentioned weaknesses.

---

### Official Review · Reviewer_gqeb · 2025-11-01

**Soundness:** 2
**Presentation:** 2
**Contribution:** 2
**Rating:** 2
**Confidence:** 5

**Summary:**

This paper proposes an Adaptive Multi-Prototype Grouping Alignment framework for unsupervised domain adaptation on multi-category 3D detection, aiming to address the challenges of inadequate cross-domain feature alignment and ambiguous foreground-background decision boundaries. They design Noise Background Hybrid Augmentation and Noise Foreground Contrastive Learning to enhance feature alignment and mitigate the impact of unreliable pseudo-labels.

**Strengths:**

1. The cross-domain issue of 3D object detection addressed in this paper is a practical challenge in real-world autonomous driving applications.
2. Based on prior research, this paper investigates the issues of insufficient cross-domain feature alignment and ambiguous foreground-background decision boundaries.

**Weaknesses:**

1. There have already been numerous works on 2D cross-domain object detection based on prototype alignment, and several studies have also emerged in the 3D domain, such as Gpa-3D and AttProto. This paper can be considered an extension of previous work, lacking fundamental innovation.
2. The proposed method was only tested on two outdated detectors (SECOND, 2018; PV-RCNN, 2020), lacking validation of its effectiveness on the latest state-of-the-art detectors.
3. The experimental section lacks cross-domain experiments, such as Waymo to Lyft, as seen in ST3D/ST3D++.
4. The experimental section lacks a comparison with previous methods based on prototype alignment.
5. The discussion regarding the number of prototypes for different categories seems counterintuitive. In theory, vehicles, being rigid bodies, should require fewer prototypes compared to non-rigid objects like pedestrians and cyclists, which exhibit a wider variety of deformations.

**Questions:**

1. It is recommended to further analyze the impact of cross-domain adaptation on different categories.
2. It is recommended to conduct a further analysis on the impact of cross-domain adaptation under different cross-domain settings.

---

### Official Review · Reviewer_m2Uy · 2025-11-01

**Soundness:** 3
**Presentation:** 3
**Contribution:** 2
**Rating:** 4
**Confidence:** 3

**Summary:**

The paper proposes a novel framework aimed at addressing the challenges of domain adaptation in 3D object detection.

The authors introduce a framework that automatically identifies and dynamically updates feature groups to achieve precise cross-domain feature alignment. This is crucial for improving the performance of 3D object detection models when faced with domain shifts.
This technique enhances the quality of pseudo labels by integrating source domain labeled instances into potentially noisy background regions. This approach helps to mitigate the noise in pseudo labels and improves model generalization.
This method leverages prototype guidance to push low-confidence features away from decision boundaries, thereby enhancing the model's ability to distinguish between foreground and background. This addresses the issue of ambiguous decision boundaries that often arise in 3D detection tasks.
The proposed methods were validated through extensive experiments across multiple datasets, demonstrating superior performance compared to existing state-of-the-art methods in various domain adaptation scenarios.
Overall, the paper presents a comprehensive approach to improve 3D object detection under domain shifts, focusing on effective feature alignment and noise management strategies

**Strengths:**

1. The framework automatically identifies and dynamically updates feature groups with different representations, allowing for effective cross-domain feature alignment. This adaptability helps in addressing the challenges posed by domain shifts
2.  It introduces two innovative techniques—Noise Background Hybrid Augmentation and Noise Foreground Contrastive Learning. These strategies enhance the quality of pseudo labels by reducing noise in both background and foreground regions, thereby improving the model's performance in distinguishing between relevant and irrelevant features
3. The framework is designed to handle multi-category scenarios effectively, overcoming limitations of existing methods that primarily focus on single-category tasks. This capability is crucial for real-world applications where multiple object categories are present

**Weaknesses:**

1. The framework may struggle with aligning features across domains due to the complex nature of point cloud data, which can be influenced by various factors such as observation angles and distances. This complexity can lead to challenges in effectively clustering features and may result in performance degradation when dealing with multi-category scenarios
2.  The model faces difficulties in distinguishing between foreground and background due to the sparse nature of point clouds from distant or occluded objects. This ambiguity can lead to misclassification of features, where high-confidence filtering mechanisms may incorrectly label positive samples as negative
3. The reliance on pseudo labels generated from the teacher model can introduce noise, particularly in regions where the model is uncertain. This noise can adversely affect the training process and the overall performance of the model, as the quality of pseudo labels is critical for effective learning

**Questions:**

1. How do you determine the optimal number of prototypes for each category? Is there a specific criterion or method used to decide this, especially in multi-category scenarios?
2. Can you elaborate on the data-driven grouping model? What specific features or metrics are used to identify and update the feature groups dynamically?
3. In the Noise Background Hybrid Augmentation strategy, how do you ensure that the introduced source instances do not introduce additional noise? What measures are in place to evaluate the quality of the pseudo labels?
4. How does your method perform when applied to domains that are significantly different from those used during training? Are there any limitations observed in such scenarios?

---

### Note · Authors · 2025-11-18

I have read and agree with the venue's withdrawal policy on behalf of myself and my co-authors.